# Sol-Gel Coatings with Azofoska Fertilizer Deposited onto Pea Seeds

**DOI:** 10.3390/polym14194119

**Published:** 2022-10-01

**Authors:** Beata Borak

**Affiliations:** Department of Mechanics, Materials and Biomedical Engineering, Faculty of Mechanical Engineering, Wroclaw University of Science and Technology, Smoluchowskiego Str. 25, 50-370 Wroclaw, Poland; beata.borak@pwr.edu.pl

**Keywords:** sol-gel method, thin films, seeds, mineral fertilizer, germination test

## Abstract

Pure silica sol obtained by hydrolysis of tetraethoxysilane and the same silica sol doped with fertilizer Azofoska were used to cover the surface of pea seeds. The surface state of the coated seeds (layer continuity, thickness, elemental composition) was studied by a scanning electron microscope (SEM) and energy dispersive X-ray (EDX) detector. Different conditions such as sol mixing method, seed immersion time, effect of diluting the sol with water, and ethanol (EtOH) were studied to obtain thin continuous coatings. The coated seeds were subjected to a germination and growth test to demonstrate that the produced SiO_2_ coating did not inhibit these processes; moreover, the presence of fertilizer in the coating structure facilitates the development of the seedling. The supply of nutrients directly to the grain’s vicinity contributes to faster germination and development of seedlings. This may give the developing plants an advantage in growth over other undesirable plant species. These activities are in the line with the trends of searching for technologies increasing yields without creating an excessive burden on the natural environment.

## 1. Introduction

Seeds are the agricultural products which represent a source of food and resource in agricultural practices. Changes in the environment: soil degradation, salinity, drought, the spread of pests and diseases, and climate change are not conducive to obtaining higher yields necessary to feed the growing human population. Alternatives and better solutions than the large amounts of fertilizers and pesticides currently used are needed to promote seeds germination and increase in agricultural production [1,2,3]. One of such solutions seems to be seed enhancement technologies, which aim to provide optimal conditions for the development of germinating grains by protecting them and providing them with nutrients. One of the most promising and efficient seed enrichment technologies is seed coating [4,5].

Coating seeds is a way of creating an artificial, made of exogenous material, coating on the surface of natural seeds that does not obscure its shape and size, and can be an alternative to maintain or improve seed quality. These sustainable, advanced farming practices are used to achieve two main purposes [4,5,6]. The first is to protect the seeds from the unfavorable conditions of the external environment of abiotic stressors such as soil salinity, drought, or detrimental temperature ranges as well as biotic stressors: insects and fungi. The seed coatings could contain also different protectants such as fungicides, pesticides, insecticides, nematicides, and herbicides which should protect the seed from diseases, threats, and competing plant species [3,7,8,9,10]. It is known that many of the commonly used protectants have a detrimental effect on the environment, hence closing them in coatings helps to limit their widespread nature and action. The second goal of the applied coatings is to improve germination and development of the seeds by providing the necessary substances: macronutrients or micronutrients directly to the seed and sprout. In this way, the deficiencies of essential elements in the soil could be compensated. The coating may act as a cage to limit the rapid release, escape, and loss of demanded substances from the soil. With the germinating process, the substances enclosed within the coating are slowly released at the root, absorbed, and conducted to the overground part of the developing sprout [11]. The coatings may also contain symbiotic organisms (rhizobia [12,13,14] and arbuscular mycorrhizal fungi [12,15]), which lead to better germination and growth of the seedlings, and mitigate soil salinity [12,13,16]. Also interesting are the seeds covered with hydrophilic materials or hydroabsorbers (hydrogels, absorbent polymers), which are characterized by a high ability to accumulate water easily available for germination and seedling development [3,17,18,19]. Coated seeds loaded with various substances have been practiced for several decades. Today, seed coating has become a common process in plant production but there are still some problems associated with it. Low coating ratio, low stability of the layer, low germination percentage of coated seeds, slow germination, high cost, and difficult safe storage are the essentials that still need a solution.

Seed coatings may be the answer to the problem of providing the growing plant with essential nutrients at an early stage of development and increasing their competitiveness against the seeds of unwanted species (e.g., weeds). Ecologically safe coatings that minimize the environmental impact of certain materials (e.g., plant protection products) by applying them as a coating directly to the grains are particularly desirable. Most seed coatings are based on biopolymers because of their eco-friendly characteristics (they are a renewable source) and production of nontoxic residues when these materials are biodegraded in soil. As a coating, the following materials were tested: starch [19], gum [20], gelatin and polyvinyl alcohol [14], cellulose derivatives [7], chitosan [7,20], amphiphilic polymers [21], dolomitic limestone and aluminum silicate [22], biomaterials based on protein [13], and others [6,23].

Another potential seed coating material is silica. However, this material has not been tested as a seed coating, it has been rather studied as a carrier for delivering active ingredients (biocides) to the plants [24,25] or germination enhancing factor [26]. Only one article can be found in which the seeds were covered with silica sol-gel coatings prepared in an environment with different pH [27]. Silica appears to be a good option for seed treatment and feeding. It is widespread in nature and it is known that a number of silicon compounds are easily absorbed by the root system of plants. In some plant species, silica creates intracellular or extracellular silica bodies (phytolites) which are necessary for growth, mechanical strength, stiffness, and protection against fungi [28]. Silica prepared under mild conditions using the sol-gel method seem to be ideal for encapsulating fertilizer ingredients, since it is possible to enclose inside them sensitive biological materials such as enzymes [29], live cells [30], bacteria [31], algae [32], and others [33]. Encapsulated materials are not damaged. A similar approach is applied in biomedical research where silica materials synthesized by sol-gel method are being investigated as potential drug delivery systems in which drug molecules are enclosed in silica particles [34]. Thus is derived the idea of transferring silica from biomedical to agricultural applications [35].

The purpose of this work was to investigate the possibility of making thin silica coatings on pea seed surface. The SiO_2_ coating was applied by immersing the seed in a suitable sol. The effect of the seed immersion time in the sol on the thickness of the formed coating was investigated. Attempts were also made to “slim down” the coating by rinsing in solvents (H_2_O or EtOH) and diluting the starting sol. A silica sol doped with Azofoska fertilizer containing typical macros and microelements necessary for plants was also prepared and deposited onto the surface of the seeds. The seeds covered with coatings of pure SiO_2_ and SiO_2_-fertilizer mixture were subjected to a germination test. This combination of SiO_2_ coatings and fertilizer provided a favorable environment for the developing sprouts.

## 2. Materials and Methods

### 2.1. Chemicals

Tetraethoxysilane-TEOS (99.9% Alfa Aesar Gmbh&Co KG, Karlsruhe, Germany), ethanol EtOH (96%, POCH, Gliwice, Poland), hydrochloric acid HCl (36–38%, POCH, Gliwice, Poland), and distilled water H_2_O were used for the SiO_2_ coatings preparation.

### 2.2. SiO_2_ Sol Synthesis

#### 2.2.1. Preparation and Deposition of Pure SiO_2_ Sol

A SiO_2_ sol was prepared by the sol-gel method. TEOS (22 mL), EtOH (9 mL), H_2_O (5 mL), and HCl (3 μL) as a catalyst were mixed together on a magnetic stirrer at room temperature for 30 min. Then, the sol was ready to use (to deposition).

The SiO_2_ coatings were deposited by the immersion of a single seed in the SiO_2_ sol at room temperature (25 ± 2 °C) for 5, 15, and 30 min. During the immersion, the sol was stirred in an ultrasonic cleaner or on a magnetic stirrer. Since the obtained coatings turned out to be thick, modifications were made to the method of applying the coatings in order to make them thinner. The modification of this process, aimed at “thinning” the layer, consisted in immersing the grain in EtOH in an ultrasonic cleaner for 5 or 10 min after SiO_2_ sol deposition. Another “slimming” modification was the use of diluted SiO_2_ sol, which was described in Section 2.2.2.

#### 2.2.2. SiO_2_ Sol Dilution

The pure SiO_2_ sol, prepared as described in Section 2.2.1, was diluted with EtOH (96%) or distilled water in the volume ratio sol:solvent of 1:4 or 1:2. The details of the samples prepared using these sols are presented in Table 1. All the coated seeds were air-dried at ambient temperature.

### 2.3. Seed Germination Test

#### 2.3.1. Preparation and Deposition of SiO_2_ Sol with Fertilizer

The SiO_2_ sol with fertilizer was applied to the seeds intended for the germination test.

As a fertilizer, the Azofoska product (GRUPA INCO, Warsaw, Poland) was used. Azofoska is a multi-component fertilizer used for fertilizing garden plants. Azofoska consists of the following macroelements: nitrogen (13.6%), tetraphosphorus decoxide (6.4%), potassium oxide (19.1%), and magnesium oxide (4.5%), and microelements: copper, manganese, iron, zinc, boron, and molybdenum.

The fertilizer Azofoska (0.0536 g) was dissolved in 5 mL of H_2_O and as an aqueous solution was added to the SiO_2_ sol prepared in the same way as presented in the Section 2.2.1. The seeds were immersed in such prepared SiO_2_-fertilizer sol, and also in pure SiO_2_ sol (for comparison) for 5 min. This is schematically shown in Figure 1.

After seed deposition with SiO_2_-fertilizer sol, one group of seeds was immersed in EtOH (96%) solution for 5 min, and the second group of seeds was not subjected to this procedure. Thanks to this, grains covered with a SiO_2_-fertilizer sol of different (thinner and thicker) thicknesses were obtained.

In the pea seeds germination test, the following samples (pea seeds) were used:(1)seeds covered with pure SiO_2_ sol;(2)seeds covered with SiO_2_-Azofoska sol and post-immersed in EtOH (thinner coating);(3)seeds covered with SiO_2_-Azofoska sol, not immersed in EtOH (thicker coating);(4)seeds without coatings as control.

After the coatings were applied, the seeds were air-dried at ambient temperature. No preheating in the dryer was used.

#### 2.3.2. Seed Weight Changes over Time

Six pea seeds were selected for the study of weight changes over time. As a reference, seeds without SiO_2_ coatings (seeds from the packet) were used. They were weighed for the next 3 days. In the case of layered seeds, they were weighed before applying the sol, 1 and 2 days after depositing the sol.

#### 2.3.3. Germination Test

The five grains of every researched group (presented in Section 2.3.1) were “seeded” in Petri dishes with cellulose wadding. Each dish was filled with 35 mL of tap water; all the dishes were stored on the windowsill at ambient temperature. During the experiment, the dishes were replenished with water. On the next days, the plates were flooded with the following portion of water: 2nd day—15 mL; 3rd day—10 mL; 4th day—15 mL; 5th day—10 mL; and 6th day—10 mL. The seeds were immersed in water all the time.

### 2.4. Pea Seeds

For the experiments, pea seeds (*Pisum sativum* L.) commercially available in Poland (Specialist Seed Farm Grzesiak—polish producer of seed materials) and intended for sowing were used. Polish name of the plant is *groch siewny cukrowy Ilówiecki*. Pea seeds were selected for the research due to the convenient size and shape of seeds and quick germination.

### 2.5. Methods

The presence of formed SiO_2_ coatings and their adhesion to the seeds’ surface was observed using a scanning electron microscope SEM S-3400N HITACHI (Hitachi High-Technologies Corporation, Tokyo, Japan) equipped with an energy dispersive X-ray (EDX) detector. The samples (coated seeds) were placed on an aluminum holder by using double-sided carbon tape intended for SEM measurements. The samples were not covered with any layer of conductive material (neither carbon nor gold) so as not to introduce an additional layer on the seeds. Measurements were made in the low vacuum mode, which allowed to protect the seeds from drying out and damage that would occur in the normal operation of the microscope in high vacuum. This technique of low vacuum mode is advantageous for the observation of specimens such as biological samples, moist samples, liquids, polymers, and ceramics [36]. The parameters selected in order to obtain a good quality image of the seeds were: energy of the primary beam 10 kV, chamber pressure of 40 Pa, and a working distance in the range 8.5–12 mm. The low vacuum mode in the microscope is combined with the backscattered electron (BSE) detector.

## 3. Results

### 3.1. SiO_2_ Coatings Deposited in Different Conditions

The first approach to obtain SiO_2_ layers on pea seeds was their immersion in the SiO_2_ sol for various periods of time: 5, 15, and 30 min. During the immersion, the sol was kept in an ultrasonic cleaner (5, 15, and 30 min) or on a magnetic stirrer (5 min). The SEM micrographs of prepared coatings are presented in Figure 2. There are micrographs with different magnifications, and the thicknesses of the layers are marked on the images.

As can be seen in the presented micrographs, the formed SiO_2_ layer is cracked and in some places, it even fell off the seed surface. Thus, the seeds are often without coating or with its remains. This effect is observed for each seed regardless of the length of time of immersion in the sol (5, 10, or 30 min) and the method of mixing (ultrasonic cleaner or magnetic stirrer). It is difficult to determine the effect of the time the seed was immersed in the sol on the thickness of the formed coatings because they do not have the same thickness over the entire surface of the tested seed. The measured thickness of the various fragments ranged from 3 to 20 μm. The pea seeds do not have a perfectly spherical shape or an even surface. For this reason, the applied layers, even if they cover the entire grain, do not have the same thickness. The seed also lives after applying the coating and over time, especially in dry conditions, the seed loses water, shrinks, and forms “wrinkles” on the surface. In some places the surface of the seed is sunken and forms pits where more sol has accumulated during the application, hence a thicker layer was then formed. Unfortunately, in these places, cracks in the coating begin, which then spread over almost the entire surface of the seed. This is clearly visible in the SEM images. The coating fragments from the well are thicker and become thinner as they come to the flat surface. In addition, the Si content (20–50 wt. %) determined by the EDX measurement varies depending on the layer thickness. Thicker sections of the coating contain more Si, and with thinner coatings, less Si.

Of course, the thickness determined from the SEM image is not perfect. The broken layer fragments lie at different angles, hence the measurement of their thickness is error prone. It is difficult to find a fragment that is oriented perfectly perpendicular to the camera. The micrographs with the marked thicknesses presented in Figure 1 showed the best orientation of the broken fragments of the coatings that have been found. It was also observed that the underside of the coating reflects the topography of the seed surface. It was visible on the pieces of coating turned upside down.

The state of the surface of the pea seeds presented in Figure 1 changes under the influence of the treatment in the SiO_2_ sols. It is caused by the formation of a coating that covered the surface relief. This confirms that the SiO_2_ coating must be deposited on the surface of the seeds.

In summarizing, the layers obtained by immersing the seeds in the sol are thick and break quickly. In order to improve the quality of these coatings, an attempt has been made to deposit a thinner layer. For this purpose, the seeds, after applying pure SiO_2_ sol (5 min immersion time in an ultrasonic cleaner or on a magnetic stirrer), were immersed in EtOH in an ultrasonic cleaner for 5 or 10 min. In our earlier studies, the coating deposited on the lentil grains and treated in this way became thinner. In the case of peas, this procedure failed because the coating was completely washed out. EDX analysis did not show the presence of Si.

Another action taken to obtain thinner layers was diluting the starting sol. The sol was diluted with EtOH (96%) or distilled water in 1:4 by volume ratios (samples 025EtOH and 025H_2_O, respectively) and 1:2 by volume ratios (samples 05EtOH and 05H_2_O, respectively). The SEM micrographs of the coatings formed with diluted sols are presented in Figure 3.

The more diluted sols created better (continuous) layers. The best layer was obtained using SiO_2_ sol diluted 1:4 by volume ratios with EtOH (025EtOH, Figure 2, first raw). The resulting layer was thin, and it reflected the texture of the seed surface. The presence of the SiO_2_ layer was confirmed by the EDX test, which indicated the presence of Si. The Si content in this layer was in the range of 8–21% by weight. This was less than the Si content in the layers obtained by immersing the grain in undiluted sol (25–38 wt. % Si). The coating obtained by diluting the sol in the same volumetric ratios but with water (025H_2_O, Figure 3, second raw) was also thinner than the coating formed from undiluted sol but heterogeneous. On one side of the seed, the layer was thicker and cracked during drying. Cracks are visible in the micrographs from SEM, some of the cracked coating came off. Content of Si in this coating was in the range 5–17 wt. %. Both kinds of coatings prepared from the sols diluted 1:2 (05EtOH and 05H_2_O) were cracked to a significant degree regardless of the used solvent (EtOH and H_2_O). Some fragments were also missing. The EDX analysis for some points of 05EtOH cracked coatings showed 41–46 wt. % Si. This part of the cracked coating was in the hollows of the seed (Figure 3, third raw), hence it was thicker and contained more Si. It was also the reason that this coating was the most fractured and lacking in continuity.

The EDX analysis for 05H_2_O showed 15–23 wt. % of Si. This coating was also cracked and showed local defects. The coating thicknesses shown in Figure 3 do not reflect the entire coating. The thickness of the cracked layer fragments was estimated at 2–6 µm. No thicker fragments could be measured as they fell off or were misoriented.

### 3.2. Seed Germination Test

For the germination test, seeds coated with a pure SiO_2_ sol and a SiO_2_ sol doped with commercial Azofoska fertilizer were selected. The coatings were prepared by immersion of the seeds in the proper sol for 5 min. After the immersion in SiO_2_-fertilizer sol, one group of seeds was immersed in EtOH (96%) solution for 5 min; the second group of seeds was not subjected to this procedure. The intention was to obtain a SiO_2_-fertilizer coating of varying thickness. The layer should be thinner after dipping in EtOH. Pea seeds from an original bag (without any artificial coating) were used as a control in this test. In summarizing, four groups of seeds were subjected to the germination test:(1)seeds covered with pure SiO_2_ sol;(2)seeds covered with SiO_2_-fertilizer sol—immersed in EtOH (thinner coating);(3)seeds covered with SiO_2_-fertilizer sol—not immersed in EtOH (thicker coating);(4)seeds without coating as control.

Each group consisted of six seeds of different sizes and weights, as presented in Figure 4. Five seeds of each group were assigned for the germination test, and one seed from each group (marked with SEM) was not subjected to the germination test as it was intended for the SEM test to see the surface of the seed (coating) before the sprouting process. Photos and SEM micrographs of the sprouted seeds are presented in Figure 5 and Figure 7, respectively, and SEM micrographs of seeds not subjected to the germination test are presented in Figure 6.

The seeds that have sprouted and developed first leaves are marked with a leaf on the charts in Figure 4 and they are clearly visible in Figure 5. The seeds that have not sprouted at all during the germination test are marked with “×” on the charts. They were generally small seeds, but this was not always the case. Also in natural conditions, not all seeds germinate either. Some do not, for various reasons. The seeds did not sprout evenly, but that is also normal. Some seeds have germinated but have very slowly developed leaves, they are marked with “+”.

The process of seed germination starts with the uptake of water by the seed through imbibition. The first sprouts appeared on the second day after sowing, but not on all seeds. The sprouts had all the seeds of the reference sample and the sample with a thicker layer of sol-fertilizer. For the remaining samples, three germinated seeds were observed. A total of 7 days after sowing, the reference sample had four seeds—plover and 1 seed with a sprout (its growth stopped at this stage). Three out of five seeds covered with the SiO_2_ layer (without fertilizer) developed seedlings, the other two did not germinate at all. The seeds covered with a thinner layer of sol-fertilizer developed three seedlings, one seed stopped at the sprout stage, and one seed did not sprout at all. The best-developed seedlings were observed for seeds covered with a thicker layer of SiO_2_-fertilizer sol. All five seeds developed pretty seedlings. During the germination test, no plant died (nor withered). The photos of the samples are presented in Figure 5.

After one week, the SiO_2_ coated seeds not intended for the germination test and the air-dried sprouted seeds were tested by SEM.

#### 3.2.1. SEM Test of the Seeds Not Subjected to a Germination Test

The pea seeds immersed in different sols (with and without fertilizer) were covered by coatings formed by the SiO_2_ network. The presence of these coatings was confirmed by SEM micrographs, and EDX analysis. The relief of the seed surface was clearly visible in the images of pure seeds (Figure 6, first row) and it became invisible under the applied SiO_2_ layer. The SEM micrographs of the pea seeds covered with SiO_2_ coatings reflects the condition of the seed prior to germination test and are presented in the Figure 6. These are the same seeds marked as SEM in Figure 4.

As is observed in the SEM micrographs, the coated layer is not continuous and does not uniformly cover the entire surface of the seed. The coatings consisted of a pure SiO_2_ layer, and a thinner SiO_2_-fertilizer layer are cracked but do not fall off the seed. The worst adhesion had the thicker layers of SiO_2_-fertilizer, which was significantly cracked and fell completely off in some places, revealing the surface of the seed.

The pure pea seed contains carbon, oxygen, magnesium, silicon, potassium, calcium, and traces of aluminum. These elements were determined by the EDX method, but they were not observed on the coated seeds. The EDX analysis confirms the presence of the SiO_2_ layer on the surface of the seeds. The amount of Si measured for the seeds coated by the SiO_2_ layer clearly increased as compared to the Si content observed for pure seed. The pure seeds had a Si content of 2–7 wt. %; SiO_2_-coated seeds had a Si content of 23 to 36 wt. %.

#### 3.2.2. Seeds after Germination

After the germination test, the seeds that developed sprouts were examined with the SEM and EDX analyses. The obtained SEM micrographs are presented in Figure 7.

The germination and development of seedlings drastically changes the seeds. The seeds break and falls apart. These changes obviously affect the outer SiO_2_ coating, which deteriorates and falls off. This effect was observed for all sprouted seeds. No shell fragments floating in the solution in a Petri dish were observed with the naked eye.

## 4. Discussion

The silica coatings pure and doped with different substance (fertilizer) could be easily produced by the sol-gel method. It is known that silica coatings show good adhesion to various materials (metallic substrate [37], textile fabrics [38], polymers [39], and others [40]), thereby the idea to apply them on seeds emerged. Reagents used in a sol-gel synthesis (silica precursor and water) as well as ethanol as a by-product, are not harmful to seeds, but the idea of using sol-gel coatings for seeds modifications is poorly presented in the literature. The thin sol-gel layers formed from silica sols with different pH values and different starting TEOS concentrations were studied as a material for barley seeds coating [27]. It was observed that seed treatment in silica sols (regardless of their pH value) enable the creation of silica coatings on the seed surface and these coatings did not adversely affect the seeds. All coated barley seeds showed a tendency toward root and sprout growth [27].

Additionally in the presented research, the sol-gel coating applied to the surface of the pea seeds does not interfere with the germination and development of the seedling. The sol-gel coated seeds showed vital activity and sprouted. The fact that some coated seeds have not germinated at all or stopped developing at the sprout stage is a natural effect. In nature, not all seeds germinate and develop into the final plant form. Some of the seeds may not be viable because they are empty (they do not have an embryo) or the embryo does not have enough growth potential to overcome the mechanical restraint of the natural seed coat and endosperm [41]. These reasons have nothing to do with the artificial coatings produced on the seeds. The sol-gel coatings deposited on the surface of the seeds did not limit the development of sprouts, which formed the first leaves within a week. Sometimes the produced coatings could be cracked and defected. It is worth mentioning that cracks in the coating are not critical, but what is more important is the adhesion of the coating to the seed surface. A crack in the layer may facilitate germination and sprout development through increased water uptake. The presence of sufficient water is important to initiate the enzyme activity and metabolism of the seed, so before the seed can germinate it must collect water from its surroundings. Natural changes occurring in the seeds during germination (water imbibition, swelling of water and gas) cause the bursting of the seed coat [42]. Additionally in the germination test presented here, the processes of water imbibition and swelling caused the total destruction of the sol-gel silica coating that cracked and detached from the surface of the seeds. It is clearly visible in the photos presented in Figure 7. Once the coating has broken, the nutrients can enter the soil and nourish the seedling. Probably thanks to this, the seedlings of seeds covered with a thicker SiO_2_ coating with fertilizer developed the most and all the seeds germinated. The coating partially peeled off, but the nutrients released in the vicinity of the seed fed the developing sprout of the pea. In the case of seeds covered with a thinner layer of gel, some of the micronutrients could be washed out during immersion in EtOH.

The main component of SiO_2_ coatings is silicon. Silicon is the second most abundant element in soil and mainly exists as silica and silicates, the forms, which cannot be absorbed by plants [43]. Plants absorb and utilize Si in the form of orthosilicic acid Si[OH]_4_, but the dissolution of Si from soil minerals is a possible but slow process [44]. Although Si has not been classified as an essential element, it is found in all terrestrial plants. Its role is indisputable. It is claimed that Si takes part in alleviating various biotic and abiotic stresses in plants (salinity, drought, and heavy metal toxicity) [43,44,45,46]. There is evidence that Si also significantly increases germination characteristics (e.g., germination percentage, germination rate, and shoot length) [43]. Thus, the sol-gel technology and prepared coatings could contribute to the improvement of seed quality.

It seems that the proper composing of the seed coating may have a significant influence on germination rate and growth of the plant at the initial phase of development. By supplying water and minerals, the coating can create almost ideal conditions for the development of the seedling, giving an advantage over other seedlings right from the start. Generally, in the tested cereals, coating significantly reduced germination rate and total germination as compared to uncoated seeds [17]. It was also studied an influence of coat thickness and composition on the germination rate. Thicker coating, larger than 75% of the total grain size, showed germination rates close to those of uncoated seed. Coating with the thickness below 50% reduced germination rates. The seed coatings also help to prevent seed ageing [21]. The artificial coating formed on the surface of the seeds provides a physical barrier to reduce the loss of nutrients as well as to reduce the diffusion of gases between the seeds and the external environment. All these effects help maintain a favourable environment inside the seeds.

The sol-gel coatings formed on the seeds surface generally do not harm them, even if in the first stages of plant development (germination) there are no significant differences from the control groups, they may appear in the later stages of plant growth. The sol obtained in the sol-gel process consists of water and alcohol (most often ethanol), so soaking the seeds in this sol hydrates them, and the presence of ethanol can provide protection against microbes.

## 5. Conclusions

The SiO_2_ sols (pure and doped with fertilizer Azofoska) were deposited on the pea seeds surface by their immersion in the proper sol mixed by an ultrasonic cleaner or magnetic stirrer. Regardless of the method of mixing and the time when the seeds were immersed in the sol, the obtained coatings turned out to be heterogeneous, quite thick (3–20 µm), and cracked. Many fragments of the coating fell off completely, exposing the surface of the seed, and in some places the coating still holds despite the present cracks. The heterogeneity in the thickness of the coatings may be due to the shape of the grains, which are not perfectly flat but contain pits. More sol accumulates in these cavities and thus, the thick layer shrinks and cracks during drying. A way to obtain thinner coating was to use sols diluted with water or EtOH. The continuous SiO_2_ layer was obtained using SiO_2_ sol diluted 1:4 with EtOH.

The germination test showed that the treatment of pea seeds in silica sols had no negative effect on the development and growth of the sprouts. The seeds covered with SiO_2_-coatings germinated at the same time as the control seeds (without coating). The SEM micrographs clearly show that all the coatings on the seeds completely cracked during the germination test, but this was due to the swelling and disintegration of the seed. The seeds covered with a thicker layer of SiO_2_ with fertilizer turned out to be the best developed one week after sowing.

Using SiO_2_ coatings with Azofoska fertilizer showed that after depositing such coatings, the seeds still show vitality and an ability to germinate. The use of such a protective and nourishing coating on the seed surface is intended to enhance its growth and development potential in order to obtain higher yields. These activities are in line with the trends of searching for agricultural technologies that increase yields and reduce the negative impact on the environment.

## Figures and Tables

**Figure 1 polymers-14-04119-f001:**
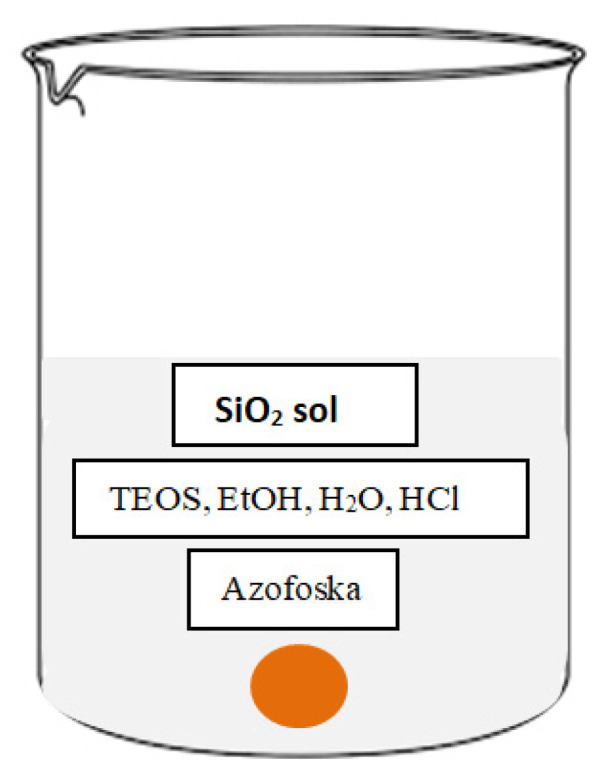
Deposition of the SiO_2_-Azofoska coating on pea seeds.

**Figure 2 polymers-14-04119-f002:**
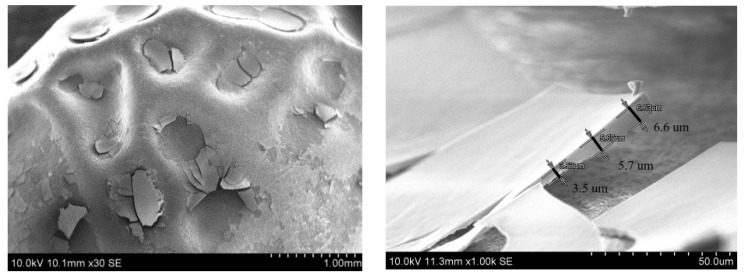
SEM micrographs of the pea seeds after different time of SiO_2_ sol deposition: 5 min—ultrasonic bath (**first raw**); 10 min—ultrasonic bath (**second raw**); 30 min—ultrasonic bath (**third raw**); and 5 min in a container on a magnetic stirrer (**fourth raw**).

**Figure 3 polymers-14-04119-f003:**
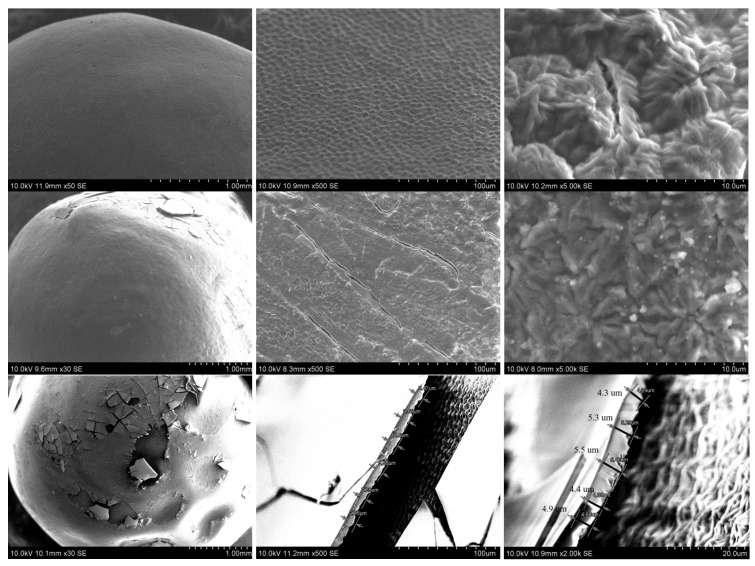
SEM micrographs of the pea seeds after diluted SiO_2_ sol deposition: 025EtOH (**first raw**), 025H_2_O (**second raw**), 05EtOH (**third raw**), and 05H_2_O (**fourth raw**).

**Figure 4 polymers-14-04119-f004:**
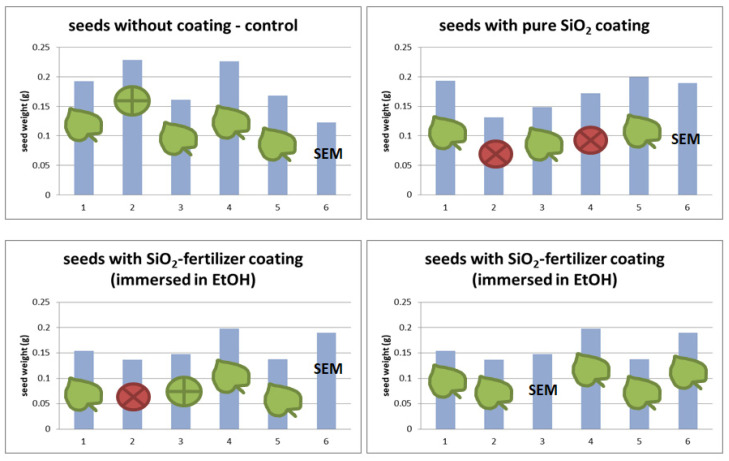
Weights of the seeds chosen for the germination test: “leaf”—seed sprouted and development; “+”—seed sprouted; “×”—seed not sprouted; “SEM”—seed intended for SEM test.

**Figure 5 polymers-14-04119-f005:**
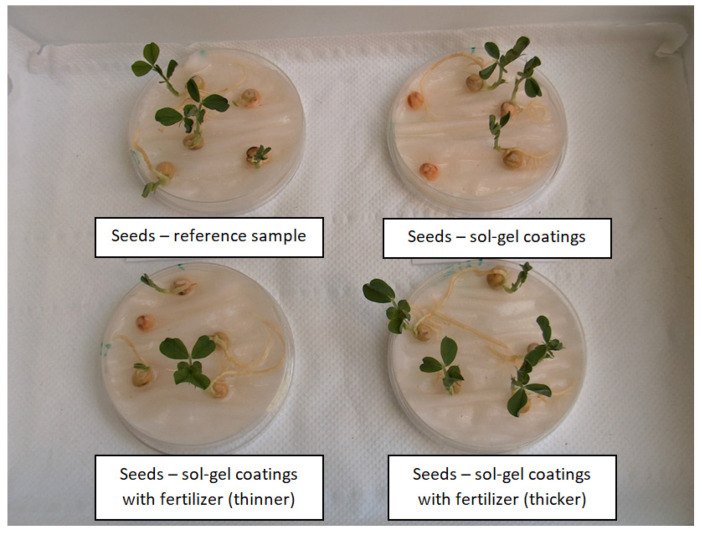
Photos of the researched seeds one week after sowing.

**Figure 6 polymers-14-04119-f006:**
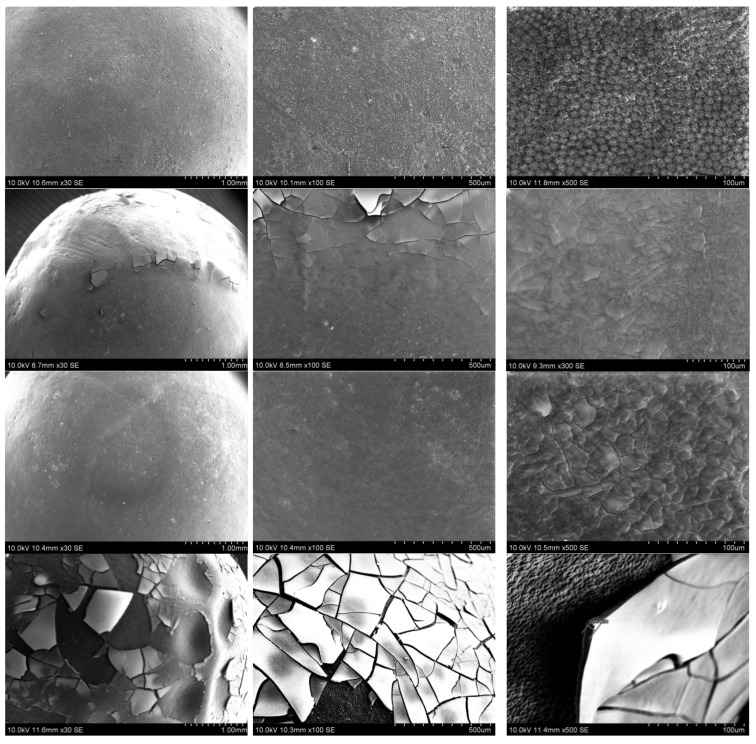
SEM micrographs of the pea seeds not subjected to a germination test: **first row**—pure seed (without SiO_2_ coating); **second row**—seed with SiO_2_ coating; **third row**—seed with thinner SiO_2_-fertilizer coating; and **fourth row**—seed with thicker SiO_2_-fertilizer coating.

**Figure 7 polymers-14-04119-f007:**
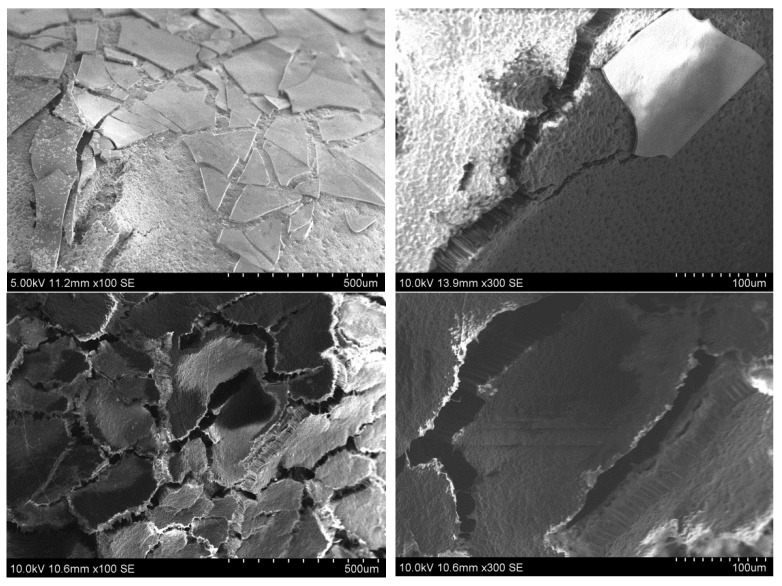
SEM micrographs of the pea seeds after germination test: **first row**—seed with pure SiO_2_ coating; **second row**—seed with thinner SiO_2_-fertilizer coating; and **third row**—seed with thicker SiO_2_-fertilizer coating.

**Table 1 polymers-14-04119-t001:** Samples of pea seeds coated with SiO_2_ sol in different conditions.

Sample	Description
5	Seed after 5 min in SiO_2_ sol in ultrasonic cleaner
15	Seed after 10 min in SiO_2_ sol in ultrasonic cleaner
30	Seed after 30 min in SiO_2_ sol in ultrasonic cleaner
5\5	Seed after 5 min in SiO_2_ sol in ultrasonic cleaner and 5 min in EtOH
5\10	Seed after 5 min in SiO_2_ sol in ultrasonic cleaner and 10 min in EtOH
5M	Seed after 5 min in SiO_2_ sol on magnetic stirrer
5M5	Seed after 5 min in SiO_2_ sol on magnetic stirrer and 5 min in EtOH
025EtOH	Seed after 5 min in SiO_2_ sol diluted with EtOH (1:4) by volume ratios
025H_2_O	Seed after 5 min in SiO_2_ sol diluted with H_2_O (1:4) by volume ratios
05EtOH	Seed after 5 min in SiO_2_ sol diluted with EtOH (1:2) by volume ratios
05H_2_O	Seed after 5 min in SiO_2_ sol diluted with H_2_O (1:2) by volume ratios

## Data Availability

The data presented in this study are available in the paper.

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
