# Peer review of "Sol-Gel Coatings with Azofoska Fertilizer Deposited onto Pea Seeds"

_polymers, 2022, doi:10.3390/polym14194119_

Round 1

Reviewer 1 Report

An interesting topic of the article and an unusual application of sol-gel technology for me personally, which is very relevant and could give an important result.

The presented review is very brief, I would like a more detailed understanding of what has been done in this area, and not just highlighting the subject of the work.

Line 85 “prepad” – should be written “prepared’?

Line 85 etc. liter must be written as capital L (mL) following SI rules.

SEM methods should be described in more detail (accelerating voltages, detectors, vacuum, etc.)

277-280 this should be moved to the experimental part.

Figure 4-5. Thus, the best germination was observed for samples with the most defective coating.

At the same time, there is no control sample with fertilizer without a sol-gel coating, so its effectiveness cannot be correctly compared.

The article presents an interesting method, but the significance and reproducibility of the results presented raise questions. At the very least, this coating does not impair or greatly impair the germination ability of the seeds and can actually protect them from any disease. Hopefully, future research will answer these questions.

Author Response

The answer is in the attachment.

Reviewer 2 Report

The article “SiO2 sol-gel coatings on pea seeds” is interesting and have an innovator character. However, the manuscript needs to be improved in some points to be considered to publication. This way, I have comments, questions and suggestions that can enhance the performance of manuscript that are enumerated below.

Title

1) As concise as the title is, it needs to be more specific, for example, put the fertilizer name in the title

Abstract

1) The key-words must be different from the title to the article gain more visibility.

2) I think the abstract must be more elaborate. It is a poor information text. An introduction for the importance of coating sea seeds must be added.

Introduction

3) The introduction section is confusing. Specially lines 66-71. It is not clear for why the author don’t explore the silica encapsulation of Azofoska in the introduction section.

4) The term ‘doped’ (line 76) seems to be erroneous for this purpose.

Experimental section

5) The author must provide the HCl concentration in section 2.1.1. Is it 37%? Or diluted?

6) The author says that the samples were not covered with a carbon layer? And what about Au? How the SEM images were made? The sample preparation needs to be provided.

Results and discussion

7) The author shows a study of SEM and EDX from the SiO2 deposition. What about the chemical interactions presents between seeds and SiO2? The author doesn’t think in made a XPS analysis to verify this interaction?

8)Why the author says the application of SiO2 on seeds emerged?

9) Why the author used TEOS? MTMOS could be applied too? Or another silica precursor? It must be discussed.

In the Reference Section: The references seem to be of poor quality. Journals of major impact factor must be appeared in the bibliographic revision.

Author Response

The answer is in the attachment.

Reviewer 3 Report

-          In Abstract, specify the applications of materials.

-          At “2. Materials and Methods” chapter, include the section with used materials, containing information about the concentration, manufacturer, city, country.

-          At 2.1.1., include the scheme for the prepared materials.

-          At 2.1.1., include the room temperature (i.e. 25 ± 2°C)

-          At 2.4., include more parameters selected for analysis

-          At Conclusions, specify the sectors of the industry where these materials can be used.

Author Response

The answer is in the attachment.

Round 2

Reviewer 2 Report

The author has been improved the work, for this I recommend the publication of manuscript in Polymers.

Reviewer 3 Report

Dear Sirs,

The manuscript was improved and it can be published in this form.